# Inline Quality Monitoring of Reverse Extruded Aluminum Parts with Cathodic Dip-Paint Coating (KTL)

**DOI:** 10.3390/s22249646

**Published:** 2022-12-09

**Authors:** Alexander Pierer, Markus Hauser, Michael Hoffmann, Martin Naumann, Thomas Wiener, Melvin Alexis Lara de León, Mattias Mende, Jiří Koziorek, Martin Dix

**Affiliations:** 1Fraunhofer Institute for Machine Tools and Forming Technology IWU, 09126 Chemnitz, Germany; 2Benseler Beschichtungen GmbH & Co. KG, 70806 Kornwestheim, Germany; 3Department of Cybernetics and Biomedical Engineering, Technical University of Ostrava (VŠB-TUO), 70833 Ostrava, Czech Republic

**Keywords:** quality control, coating, extrusion, failure, neural networks

## Abstract

Perfectly coated surfaces are an essential quality feature in the automotive and consumer goods industries. They are the result of an optimized, controlled coating process. Because entire assemblies could be rejected if Out-of-Specification (OOS) parts are installed, this has a severe economic impact. This paper presents a novel, line-integrated multi-camera system with intelligent algorithms for anomaly detection on small KTL-coated aluminum parts. The system also aims to automatize the previously used human inspection to a sophisticated and automated vision system that efficiently detects defects and anomalies on coated parts.

## 1. Introduction

Perfectly coated surfaces are required from many industrial and productive sectors (e.g., automotive and white goods); they result from an optimized, controlled coating process. They must be characterized as accurately as possible and with a temporal and spatial resolution for an optimum coating process. Because the coating processes are often multi-stage and the parts are further assembled in the value chain, coating defects in individual components can lead to the rejection of entire assemblies, which has severe economic consequences. In particular, the coating thickness and the surface structure are subject to many disturbing influences. For this reason, quality control must be integrated into various process steps, e.g., through coating thickness measurements, a color tone gloss analysis, or a detailed surface inspection [1].

The scope of this work is the series-production capable and process-reliable inline automated optical inspection (AOI) system for the quality monitoring of extruded aluminum parts with a black cathodic dip coating (KTL). An automatic inline inspection system was developed and integrated into a production line at the Benseler facility in Kornwestheim. The parts are about 10 mm high and have a diameter of around 15 mm. The proposed inline monitoring system and data analysis methods are intended to increase the automated, sensor-based quality control accuracy and reliability. The system complements the existing visual inspection to reduce the inspection slip quota at the end of the line and to facilitate the inspection.

## 2. State of the Art

### 2.1. Coating Technology

In principle, the issue of quality control is crucial for all visible and corrosion-affected surfaces of industrial-produced components. Due to the vast variability of part geometries, and the technical requirements regarding the coating, a typical coating can hardly be defined. Widely applied in the automotive sector are multi-layer structures of KTL primer, filler (optional), color paint, and clear coating on pretreated adhesion-activated metal substrates. Commonly, each layer is in the range of 15 to 30 µm, forming a total layer thickness of around 50 up to 80 µm [2] (p. 38). In the corrosion protection of floor assemblies or chassis, a thicker KTL coating layer is commonly applied depending on the customer requirements (e.g., 30 to 60 µm). The exact chemical composition and coating technology parameters themselves are usually part of the confidential know-how of the industrial applicant. Pigments can consist of almost any conceivable polymer, e.g., polyurethane, polyacrylate, or epoxy resins. KTL and powder coatings are deposited onto a metallic substrate with an electric field.

For this reason, the coating molecules have functional groups as ion formers or to yield a dipole character [2] (pp. 41–42). Color coating is often applied as wet paint (water based or with solvent) or as powder coating. A clear coat is sprayed on as wet paint. Therefore, a polar character of the pigment molecules is not mandatory, and one- and two-component systems are commonly used.

### 2.2. Quality Control

The VDA 16 standard summarizes the perceivable critical quality parameters of the visible surface and testing conditions for visual quality control in the automotive sector [1]. In addition to the boundary conditions for process line integration and automatable handling and inspection processes, VDMA 2632 summarizes the evaluation criteria for inspection systems: measurement uncertainty, detection rate, classification rate, false positive/negative rate, repeatability, long-term stability, transferability, measuring equipment capability, and measurement process capability [3]. 

Only spot-measuring systems exist for the determination of the hue and gloss of coatings, which do not allow high-rate, process-integrated measurements in large inspection fields. Similar limitations in terms of the process integrability are encountered in coating thickness measurement and deflectometry systems for a coating structure analysis. The current solutions for a non-destructive coating thickness measurement based on inductive principles, the terahertz [4] or X-ray fluorescence analysis [5], detect unique measuring spots, which must have direct contact with the component to be tested or can only be carried out in a laboratory [6]. Pulse or lock-in thermography methods can measure the coating thickness in a limited area. However, they are sensitive to interfering influences, and high-reflecting surfaces inhibit the coupling of the thermal excitation. The stationary and robot-based deflectometry systems that detect coating structure defects are costly and too slow for fast-running processes [7]. In general, robot-based systems require a precisely known shape and position of the parts to be inspected and are comparatively slow, which is an obstacle to inline integration. Another promising approach to detecting coating defects is shape-from-shading systems for the fusion of images of the same region from different illumination angles to highlight minor topographic defects [8]. Within recent years, besides classical image processing chains, the utilization of supervised learning deep neural networks for finding trained defect patterns and unsupervised learning methods based on generative neural networks emerged. The variation in the placement tolerances and illumination, as well as the limited availability of representative training datasets, reduces the reliability of such approaches [9]. Moreover, the limited adjustability for the operators are considerable obstacles for the integration of such approaches into industrial processes. 

It should be noted that automated, detailed quality monitoring is hardly established in rapid small-part production with cycle times in the range of or below a few seconds per part, and such processes can so far only be monitored on a random basis.

### 2.3. Light Propagation in Paints

The effect of the transmission of radiation has been known since the early days of the development of synthetic coatings. Thereby, the Kubelka–Munk theory [10,11] approximates the light propagation in the coating layer as scattering and an absorption model for the incoming light ray from the surface and the reflected ray backward from the substrate to the surface in high absorbing materials, such as non-transparent coatings. 

The absorption and scattering coefficients depend on the wavelength, with short-wave wavelengths being absorbed more strongly. As a result, long-wave light reflected from the substrate increasingly penetrates back to the surface at low coating thicknesses, which can cause a slight red discoloration in thin coatings. In [12], this effect was used to calibrate a hyperspectral sensor for a specific coating to measure the coating thickness from the observed changes in the backscattered spectra. The sensor must be placed directly onto the coating surface, which limits the integrateablity into production processes.

### 2.4. Color Spaces

Color is a physiological perception phenomenon created by triggering the short, medium, and long wavelength cone cells of the retina and finally creating a stimulus corresponding to a color sensation known to the brain by the incidence of a visible electromagnetic wave into the eye. Physiologically, the cones are interconnected to the channels of the known complementary colors blue-yellow and red-green and also contribute to the brightness stimulus. The main brightness stimulus is generated by the strongly photosensitive rod cells. The CIE 1931 color-space models describe the spectral tri-stimulus sensitivities of the cones of an average human [13]. For the coating industry, the L*a*b*-space [14] was derived from the CIE space. The L*a*b*-space is widely applied in the automotive sector [15] (pp. 74–77) because it models the human red-green and blue-yellow system and reproduces the non-linear human perception of color distances (MacAdam ellipses). 

The HS∗-models were developed to represent human intuitive terms to describe colors. Humans express the hues *H,* independent of brightness. Humans perceive the ratio of the white portion mixed with the pure hue (e.g., “light red” or “dark red”), which denotes the saturation or purity *S* of the color [16]. *H* and *S* are used in a mathematically similar way in all HS∗-models despite small differences in normalization [17]. The third component (***) in the respective HS∗-models represents gray tones from black to white in the model-specific scaling. The *-values differ as follows: *I* stands for “Intensity”, *V* for “Value”, *B* for “Brightness”, and *L* for “Lightness”. The advantage of the HS∗-color models is that they can determine the hue based only on the *H* component independently of the gray value. The linear characteristic of the hue *H* to the dominant wavelength λ of the perceived color can be used to approximately evaluate the spectral changes in the appearance of objects in the visible range according to Equation (1).
(1)H=(650 nm−λ)·1.08 °/nm with 400 nm≤λ≤650 nm

In this work, the HSI color space with its transformation from the red-green-blue (RGB) color space of an industrial camera with a Bayer array is further used [18]. It should be noted that the perceived color depends on the arrangement and spectral composition of the lighting due to metamerism [19]. Therefore, standardized illuminations such as D65, with a defined color temperature and color rendering index (CRI), are used for standardized color measurements [1] (pp. 11–12). Alternatively, for comparative measurements, other illuminations are sufficient if the illumination conditions and white balancing parameter of the camera remain constant between measurements.

## 3. Development of Multi-Camera Inline Monitoring Systems

### 3.1. Requirements for AOI System

The starting point for the development of the AOI system is a specification and limit pattern catalog. In addition to unambiguous characteristics for good parts and defects, this catalog also defines limit patterns (parts with characteristics or defects that are just accepted or no longer accepted by different human inspectors) and transition patterns (parts for which different human inspectors rate differently regarding the quality criteria). The limit pattern catalog contains specifications and example images of the following defect types: (1) inclusions, (2) coating thickness defects and visible substrate material, (3) craters, (4) coating bubbles, (5) orange skin, (6) various material damages, such as scratches, notches, grinding grooves, etc., greater than 0.5mm, (7) stains, (8) throw-ups, (9) blank spots on a mantle surface, and (10) the wrong part type.

The mentioned 10 defect types shall be detected attributively on each part and subsumed as anomalies. An assignment to the individual defect classes does not take place. As the acceptance criterion for the inspection system, a confusion table was defined with a maximum pseudo scrap rate (false positive) of 3000 ppm for an unambiguous defect and limit defect patterns and a maximum inspection slip (false negative) of 0 ppm for the unambiguous defect patterns.

### 3.2. Hardware of Inline Monitoring Systems 

The whole mechanical construction of the AOI system is based on an anti-static and silicon-free aluminum frame construction and conveyor belt system with an integrated drive unit as shown in Figure 1. The camera-illumination setup was optimized to have full coverage of the region of interest and occultation avoidance based on the simulation software for the coverage analysis for moving parts, developed by Gjakova et al. [20]. Figure 2 shows the user interface.

After the KTL coating, the 198 parts are placed on transparent transport trays. Using 19 cameras, around 186 images in Full HD resolution were taken within 3 to 4 s during the continuous movement of the tray at 18 m/min. To avoid motion blur and environmental light inference, the cameras (manufacturer The Imaging Source Europe GmbH, Bremen, Germany) and illuminations (manufacturer iIM AG, Suhl, Germany) are triggered with 10 µs exposure times and a global shutter. 

The captured images are processed by the software developed for Windows systems on a 14 CPU-core system and partially on the GPU (14Core Intel E5-2690v4 2.6 GHz, 32 GB RAM, NVIDIA GeForce GT 710 1 GB GDDR3). To evaluate all the parts, 320 Mbyte of image data must be acquired and processed within the production cycle of 24 s. Due to the lack of a real-time capability of Windows systems, an additional real-time controller is required to trigger the cameras and the associated illuminations at defined tray positions. This additional real-time controller is based on a Controllino Mega hardware. To measure the part position, a rotary encoder and light barrier are attached to the conveyor belt and connected to the real-time controller as inputs. For ensuring the real-time capability, the trigger sequence is handled within the interrupt routine for the decoding of the encoder. 

At each triggering, each camera captures a group of parts (usually 4 × 4 parts). For each part, 10 views are cropped from these images: 4 showing the mantle surface (camera group 1), 4 showing the front surface at different deflectometric observation angles (camera group 2), as well as 1 top view and 1 RGB image to detect the reduced coating thickness based on the HSI evaluation (camera group 3). 

After the inline inspection station, a robot picks out the failure parts from the tray and automatically refills the tray with good quality parts. Subsequently, the tray is sent to the human inspectors, who carry out the final inspection.

### 3.3. Software of Inline Monitoring Systems 

The machine software uses the XEIDANA^®^ framework (Fraunhofer Institute for Machine Tools and Forming Technology IWU, Chemnitz, Germany) developed by the Fraunhofer IWU [21]. The framework is a visual programming system and runtime for complex data processing flows and enables optimal utilization multi-core systems. The data flows are handled as networks of interconnected modules with encapsulated data processing operations. Each module creates at least one dedicated CPU thread in which the corresponding algorithms are processed. This allows the simultaneous preprocessing of different sensor signals before the signals are fed to a common execution. In addition, the modules connected as a daisy chain can be processed in the so-called pipeline mode. This means that while one module is still calculating the output for an incoming data packet, its predecessor module can already process a new data packet at the same time. Because XEIDANA supports massively parallel processing, deep learning techniques can also be applied for the monitoring of components whose inspection criteria are not clearly specified.

Table 1 summarizes the main data processing steps during the monitoring. Due to the modular concept of the system, the classical filter-based image processing chain (Section 4.1) can alternatively be replaced by machine learning methods (Section 4.2).
sensors-22-09646-t001_Table 1Table 1Main data processing steps.1Image acquisition, including lens distortion correction.2Cropping of image patches for each part from different perspectives.3Pose correction—Due to clearance fit of the parts on the tray, the parts position can vary ±1 mm. The software compensates this deviation by applying a Euclidean transformation matrix to the patch using an enhanced correlation-based image registration algorithm [22].4Failure detection modules (for each patch)4.1 Classical filter-based vision methods4.2 Machine learning methodsDetector application applied to masks for dedicated part areas:-Pixel-wise image comparison.-Gabor-filter for finding small edges, such as scratches or pickles [23].-Windowed 2D-variance filter to detect granular defects, such as orange skin.-HSI evaluation of coating thickness.Anomaly quantification analysis (AQA).Comparison/thresholding to reference image of good-state parts.Morphology operations andblob analysis of detector outputs.5Generation of failure output results.6Protocolling of results, visualization, and outputting handshake signals to sorting robot. 

## 4. Experimental Section

### 4.1. Classical Filter-Based Machine Vision Methods

During the process-introduction phase of the system, the filter parameters were manually optimized until the detection results were in acceptable accordance with the human inspectors working at the line and the specified limit sample catalogue. Therefore, approximately 50 trays (with approx. 9.900 parts) for three different part-type generations were tested and iteratively optimized. Figure 3 shows representative detection results.

Regarding acceptance tests, it must be noted that trays tested by human inspectors may also contain transition samples. Due to the transition samples, the ground-truth data are also subject to a considerable uncertainty *s*_0_. In consequence, the total inspection uncertainty *s* to be evaluated within the scope of an acceptance test is the geometric mean value of the uncertainties of the ground-truth data *s*_0_ and the inherent uncertainties of the AOI system *s_AOI_* as shown in the following equation.
(2)$$s=\sqrt{s_{0}^{2}+s_{AOI}^{2}}$$

Under the boundary conditions of a running industrial production, the uncertainty of the ground-truth data can hardly be determined under consideration of transition patterns, because this requires extensive and time-consuming repeat tests with changing human inspectors. Therefore, the acceptance test was limited to unambiguous samples for which an uncertainty *s*_0_ of 0 ppm can be assumed. After passing the acceptance test, the AOI system was successfully integrated into the production process.

Due to new production requirements for some batches, the roughness increased in the extruded base body. Moreover, powder coating partially replaced the KTL. In particular, the increased roughness led to a higher granularity of the surface and consequently increased the false positive rate (pseudo scrap) to over 20%. To reduce the sensitivity of the orange skin and edge detectors on the front surface and the front surface edge, a batch-dependent readjustment of the parameters was necessary. The optimization could be successfully completed within half production shift with several batch changes, including an observation phase of approximately 3 h. 

In the application case, the RGB camera was used to detect red shifts of the coating caused by low coating thicknesses. Therefore, the average hue $\bar{H}$ and saturation $\bar{S}$ on the part’s front surface is evaluated in the HSI color space. The calculation of a mean angle $\bar{H}$ on the hue circle is problematic at the transition from the purple line (270° to 360°) to the red hue (0°). A possible solution is the interpretation of hue angles $H_{i}$ as polar coordinates on the unit circle and back calculation of the average hue $\bar{H}$ according to Equation (3).
(3)$$\bar{H}=atan2\left(\bar{y},\bar{x}\right)\;\;\;with\;\;\;{\left(\begin{array}{cc} \bar{x} & \bar{y} \end{array}\right)}^{T}=\frac{1}{n}\cdot \sum_{i=1}^{n}\left(\begin{array}{c} cos\left(H_{i}\right)\\ sin\left(H_{i}\right) \end{array}\right)$$

Figure 4 shows a part with reduced coating thickness among sufficient coated parts. The red shift can slightly be seen already in the RGB image. In the hue image, the signal distance is increased and decoupled from the brightness, allowing defective parts to be clearly distinguished from the good parts. The saturation *S* did not provide any useful information regarding coating thickness defects due to the location of the image pixels near the black singularity of the HSI model. For non-black paint, the evaluation of saturation should be further investigated. Furthermore, ongoing research activities of the authors deal with the calibration of the hue shift or the backscatter intensity in the infrared range with the coating thickness.

### 4.2. Machine Learning Methods 

After the commissioning of the project, new approaches are being developed. The new approaches aim to find families of anomalies (clusters), and they also aim to solve the same task but with different technologies, such as spectrometric, neural networks, among others. Currently, pixel-wise neural networks alongside feature extraction algorithms are one of the first approaches being developed. The developed method is called anomaly quantification analysis (AQA). Even though the AQA has yet to be fully finished and is yet tested only on laboratory scale outside the running production process, it will be presented and explained as part of further studies.

The idea of the anomaly quantification analysis is the arrangement of imperfections and defects based on three features: (1) their shape complexity *C* [12], (2) their size (quantitate of pixels) *QTT* which is the relative size of the anomaly with respect to the image size in percent, and (3) their texture *TT* via gray-level co-occurrence matrix (GLCM) [24].

The shape complexity (C) quantifies the randomness of a perimeter. There are several methods that calculate such features; the one being used measures the distances between each vertex which is then compared with their minimum possible distance.

The idea of AQA is to determine how anomalies interact according to the mentioned features. The current usage of the analysis is constrained to the given illumination arrangement. 

Figure 5 depicts the AQA process and the topology of the customized pixel-wise convolutional neural network (CNN) used. The green part of the AQA process topology shows how the “reference graph” is created. Initially, images with defects should be selected, because those images will be the ones creating the reference region used to compare the rest of the dataset. The user should label the defects, and then a neural network model, based on CNN layers, is used to detect the anomalies.

The CNN has a structure similar to the auto-encoders, but the bottleneck is not used as reduced version of the input image. The main difference between the autoencoder and a pixel-wise CNN model is that the optimization uses a binary map as the ground truth, whereas the autoencoder uses the same input. The input layer is a grayscale image, and the output layer is a binary map of 256 × 256 pixels each. The predicted mask from the CNN was converted into a polygon to further extract all the contours’ complexity *C*, size *QTT*, and texture *TT*. The CNN was trained based on selected images from a dataset with manually classified defective parts. In total, 259 images were selected from the dataset, and due to augmentation, 5800 additional images were created for the CNN training. Figure 6 depicts example of the used augmentations.

Thereby, the following augmentation and combinations of it were used to simulate scenarios *AP* (=augmented pixel) which are difficult to find while collecting real data *OP* (=original pixel):

*Gamma* contrast gamma by applying to the pixel values of the original image *OP*:(4)$$AP=255{\left(\frac{OP}{255}\right)}^{gamma}$$Linear contrast modifies to *OP* by applying an *alpha* blending:(5)AP=127+alpha(OP−127)Rotation augmentation changing the pixel-coordinated *OP* by rotating them around the image center. 

The dataset was split into three parts: 70% of the data for training the model, 20% for testing the model, and the 10 left for evaluation purposes. A total of 600 random images were taken from these partitions to create the confusion matrix, see Figure 7. Because the model’s output is a binary map, the classes express the probability for pixels to belong or not to defects; therefore, the OK/OK and NOK/NOK represents the number of labeled pixels correctly classified. The accuracy of the model and loss/cost values during the training process are also shown in Figure 7.

Following the AQA process, the features for quantity *QTT*, complexity *C*, and texture *TT* were calculated based on the mask extracted from the CNN output.

Figure 8 shows on the left a representative detection result of the trained CNN model during the subsequent testing phase. Input image refers to image to be analyzed. CNN-output mask is the binary map which outputs the CNN after the inference process. Feature calculation refers to the conversion of the binary map into a polygon for computing the AQA features and metrics. Figure 8 shows on the right the AQA reference graph for the tested samples, whereas a point in the formed parameter space represents a detected defect. Another use of the graph is to find clusters and classes within the anomalies. However, with the current features, we could not see clear clusters, meaning that the current features did not group the defects in clear classes in this application case. Most defects used as references were grouped in the zone pointed out by the blue arrow. 

## 5. Discussion 

Currently, the machine learning process extracts three parameters: the complexity of the perimeter, the relative size of the anomaly, and the texture of the anomaly (GLCM). The existing ideas are the addition of pixels over and under saturation, brightness, compactness, impact of colors (RGB format), and polygon characteristics. These new features are currently under development, and the findings will be presented in subsequent publications.

An open issue in the evaluation of AOI systems is the uncertainty in the ground-truth data. To also consider transition patterns for the objective evaluation criteria of AOI systems, further investigations are required in conjunction with a manual inspection. For logistical and production-related reasons, the evaluation of the AQA approach could not yet be carried out under the boundary conditions of industrial production. The comparative evaluation of the classical and the AQA approach will therefore be the subject of further investigations.

## 6. Conclusions

The proposed hardware system and methods can be used to inspect small, coated parts in large quantities to increase the quality inspection’s reliability and reduce or eliminate the human inspection. In addition, it is also a significant step forward in the digitalization of coating processes. By combining classical filter-based image processing with machine learning techniques, the system identifies the known defect patterns and detects previously unknown anomalies in the coating under realistic production requirements. 

Further work will improve the fast coating thickness evaluation with the implementation of RGB, infrared cameras, hyperspectral imaging, and the inclusion of more features in the machine learning approach. 

## Figures and Tables

**Figure 1 sensors-22-09646-f001:**
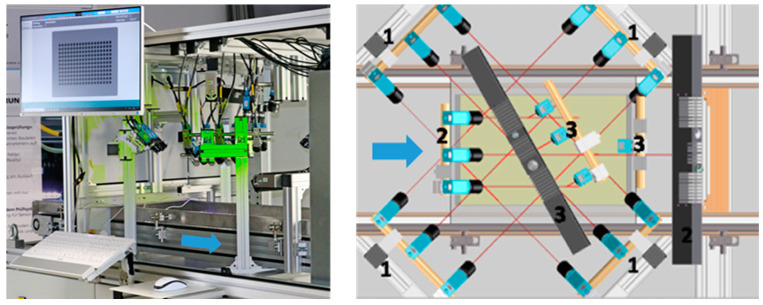
Inspection system with removed front cover plate (**left**); top view on the camera groups 1, 2, and 3 and illuminations (**right**), blue arrow—movement direction.

**Figure 2 sensors-22-09646-f002:**
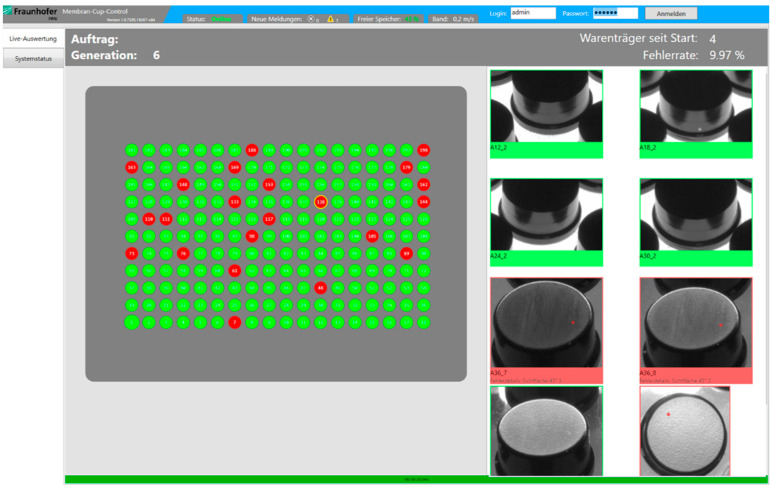
User interface of the inspection system with tray overview on the left and detailed view of a selected part on the tray (green = good status, red = defective) on the right. The results are achieved with the classical filter-based vision methods described in Table 1, Section 4.1.

**Figure 3 sensors-22-09646-f003:**
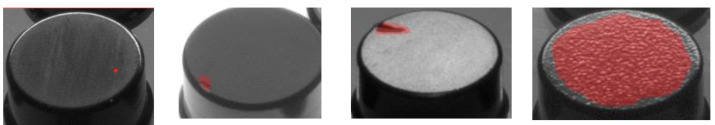
Examples for detected failures based on filter-based image processing (form left to right: inclusion, notch, notch, orange skin—all on front surface).

**Figure 4 sensors-22-09646-f004:**
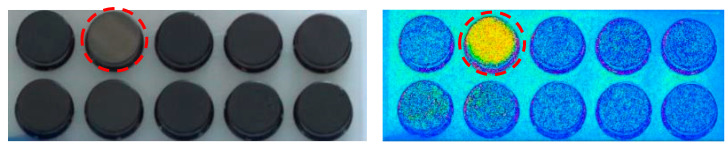
RGB image of coated parts (**left**) and hue image (**right**); the defective part (dotted circle) has a reduced coating thickness than the others.

**Figure 5 sensors-22-09646-f005:**
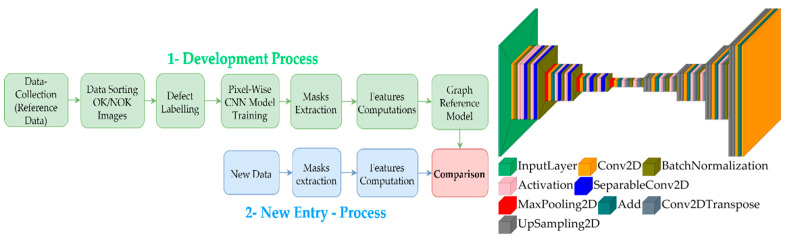
Flowchart of the AQA process (**left**); topology of CNN for anomaly detection (**right**).

**Figure 6 sensors-22-09646-f006:**
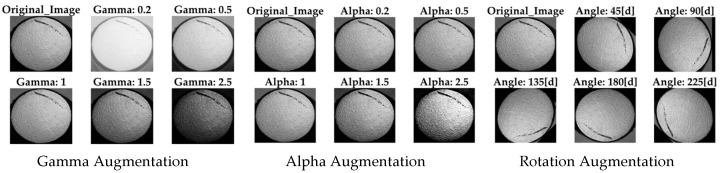
Examples for data augmentation.

**Figure 7 sensors-22-09646-f007:**
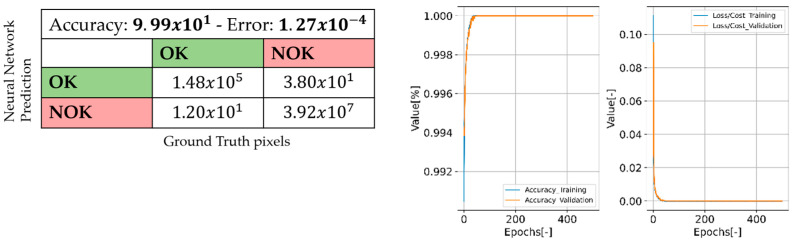
Confusion matrix (validation data), results of the training process.

**Figure 8 sensors-22-09646-f008:**
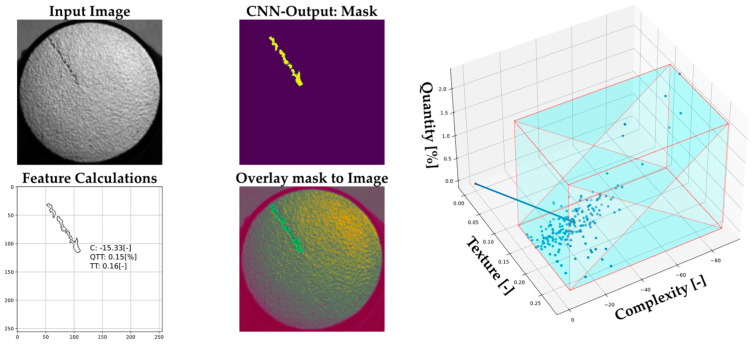
Representative results (**left**); AQA reference graph (**right**). In the right subfigure, the blue box represents the zone used as a reference for new incoming data. The red lines are the borders.

## Data Availability

Not applicable.

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
