# Peer review of "Inline Quality Monitoring of Reverse Extruded Aluminum Parts with Cathodic Dip-Paint Coating (KTL)"

_sensors, 2022, doi:10.3390/s22249646_

Round 1

Reviewer 1 Report

1. The paper introduces a novel. line-integrated multi-camera system with intelligent algorithms for anomaly detection on small KTL-coated aluminum  With parts. The system is also designed to automate the manual inspection previously used into a sophisticated automated vision system that effectively detects defects and anomalies in coated parts. The method described in this paper is more of the existing methods, only as a part of the process of linear detection, it is suggested that the author introduce their own method in more details;

2. The paper has too much introduction and too little experiment, which is a bit top-heavy. It is suggested to add the experiment part.

3. The experiment in the fourth part has a small sample size for various defects and a short observation time, so it is suggested to increase the sample size to improve the observation time to enhance the credibility of the experiment;

4. There is little introduction to machine learning, so it is suggested to introduce the specific operation process in more detail.

5. The experiment in this paper is less compared with the previous experiment, so the optimization degree and innovation degree of the algorithm cannot be clearly known. It is suggested to increase the comparison experiment to reflect the optimization degree.

Reviewer 2 Report

Inline automatic inspection is an important element of Industry 4.0. The authors present a relatively common approach applied to the false detection of coated samples using multiple-cameras array and based on classical image processing approaches (e.g. Gabor filter, color-space analysis) or popular machine learning. The chosen approaches seem adequate for the current process under investigation (extruded aluminium cylinders with cathodic dip-paint coating).  I ask the authors to address  the following points for clarification/paper's overall improvement:

line 23: please rephrase the sentence starting with "Since the coating process". It is unclear to me.

line 108: "processe".

line 116: The sentence started with "The CIE 1931..." does not contain a verb

line 135-136: reconsider the sentence. It is not clear to me

Figure 2. I can see that red/green labelling of the images is made. Which method in table 1 was used for this classification?

line 227: "evaluable"

Section 4.2: why not using a confusion matrix to summarise the ML approach? How does it correspond with the classical approach (section 4.1)?

In my point of view, Section 5 (Discussion) is not a discussion but a short conclusion. Also, it is not clear to me at all how the 2 approaches are merged or used in the real case. Do they complete each other? You mention that their combination makes the identification of anomalies possible. But how?  I suggest you add a real new Discussion section to develop your thought before mentioning it in the conclusion.

Round 2

Reviewer 1 Report

The author has revised it, it is acceptable.